# In-situ visualization of the space-charge-layer effect on interfacial lithium-ion transport in all-solid-state batteries

Longlong Wang [1,8], Ruicong Xie[2,8], Bingbing Chen[3,8], Xinrun Yu[1,8], Jun Ma [1✉], Chao Li [2✉], Zhiwei Hu [4], Xingwei Sun[1], Chengjun Xu[3], Shanmu Dong[1], Ting-Shan Chan [5], Jun Luo [2✉], Guanglei Cui [1,6✉] & Liquan Chen[7]

The space charge layer (SCL) is generally considered one of the origins of the sluggish interfacial lithium-ion transport in all-solid-state lithium-ion batteries (ASSLIBs). However, in-situ visualization of the SCL effect on the interfacial lithium-ion transport in sulfide-based ASSLIBs is still a great challenge. Here, we directly observe the electrode/electrolyte interface lithium-ion accumulation resulting from the SCL by investigating the net-charge-density distribution across the high-voltage $LiCoO_2$/argyrodite $Li_6PS_5Cl$ interface using the in-situ differential phase contrast scanning transmission electron microscopy (DPC-STEM) technique. Moreover, we further demonstrate a built-in electric field and chemical potential coupling strategy to reduce the SCL formation and boost lithium-ion transport across the electrode/electrolyte interface by the in-situ DPC-STEM technique and finite element method simulations. Our findings will strikingly advance the fundamental scientific understanding of the SCL mechanism in ASSLIBs and shed light on rational electrode/electrolyte interface design for high-rate performance ASSLIBs.

[1] Qingdao Industrial Energy Storage Research Institute, Qingdao Institute of Bioenergy and Bioprocess Technology, Chinese Academy of Sciences, Qingdao 266101, China. [2] Center for Electron Microscopy and Tianjin Key Lab of Advanced Functional Porous Materials, Institute for New Energy Materials and Low-Carbon Technologies, School of Materials Science and Engineering, Tianjin University of Technology, Tianjin 300384, China. [3] School of Energy Science and Engineering, Nanjing Tech University, Nanjing 210000, China. [4] Max Plank Institute for Chemical Physics of Solids, Nothnitzer Strasse 40, D-01187 Dresden, Germany. [5] National Synchrotron Radiation Research Center, Hsinchu, Taiwan 30076, Republic of China. [6] Center of Materials Science and Optoelectronics Engineering, University of Chinese Academy of Sciences, Beijing 100049, China. [7] Key Laboratory for Renewable Energy, Beijing Key Laboratory for New Energy Materials and Devices, Beijing National Laboratory for Condensed Matter Physics, Institute of Physics, Chinese Academy of Sciences, Beijing 100190, China. [8] These authors contributed equally: Longlong Wang, Ruicong Xie, Bingbing Chen, Xinrun Yu. ✉email: majun@qibebt.ac.cn; chao_li@tjut.edu.cn; jluo@tjut.edu.cn; cuigl@qibebt.ac.cn

All-solid-state lithium-ion batteries (ASSLIBs) have been considered one of the most promising alternatives to conventional LIBs in terms of their superior safety and great potential to meet the requirements of high energy and power density[1–5]. As an essential component of ASSLIBs, several state-of-the-art sulfide solid-state electrolytes (SEs) have achieved a high room-temperature ion conductivity of $10^{-2}\,S\,cm^{-1}$[6–11], which is close to or even exceeds that of liquid electrolytes (LEs). Nevertheless, the performance of ASSLIBs based on these electrolytes is still inferior to that of commercially available LIBs[12] because fast solid electrode/electrolyte interfacial lithium-ion transport remains a vital challenge in ASSLIBs[13–16].

The sluggish lithium-ion transport across the solid electrode/ electrolyte interface mainly results from three aspects: the space charge layer (SCL), interface reaction generating ionically resistive products, and poor physical contact. Recently, important progress has been achieved in solving the interface reaction and physical contact issues by coating, thermal soldering, or forming epitaxial interfaces[17–19]. Unfortunately, inspiring solutions for the SCL issue still remain to be explored owing to the unclear action mechanism of the SCL on interfacial lithium-ion transport in ASSLIBs. Although previous studies have tried to visualize the ionic and potential profiles in the SCL via in situ electron-holography transmission electron microscopy (EH-TEM)[20], spatially resolved electron energy-loss spectroscopy (SR-EELS)[21], and Kelvin probe force microscopy (KPFM)[22], the SCL effect on interfacial lithium-ion transport is still unclear due to the lack of direct experimental evidence of the interfacial charge distribution and accumulation[23,24]. Furthermore, it has been reported that the oxide/sulfide interface exhibits more severe SCL effects than the oxide/oxide interface[25]. However, to the best of our knowledge, an SCL visualization study related to sulfide SEs has not been reported because they are easily damaged by the electron beam, which, in turn, hinders the development of a rational interface design strategy to solve the SCL issue in promising sulfide-based ASSLIBs.

Recently, the segmented-detector differential phase contrast STEM (DPC-STEM) technique was used to reconstruct an electric field vector map and a charge-density map with higher spatial resolution than and without the restriction of specimen geometry imposed by EH-TEM[26], offering a new method to solve this challenging issue in ASSLIBs[23,24,27–29]. Here, we directly observe the interface lithium-ion accumulation resulting from the SCL by investigating the net-charge-density distribution across the electrode/electrolyte interface of a working sulfide-based ASSLIB using the in situ DPC-STEM technique. To exclude the influence of the interface reaction and poor contact on lithium-ion transport, we rationally design high-voltage $LiCoO_2$ (LCO)/argyrodite $Li_6PS_5Cl$ (LPSCl)/In-Li ASSLIBs with high cathode/electrolyte interface stability and good contact (Supplementary Figs. 1 and 2)[30,31]. More importantly, we further demonstrate a built-in electric field and chemical potential coupling strategy to reduce the SCL effect and boost lithium-ion transport across the electrode/electrolyte interface in sulfide-based ASSLIBs by the in situ DPC-STEM technique and finite element method (FEM) simulations. Our results on in situ visualization of the SCL effect on interfacial lithium-ion transport in ASSLIBs are expected to strikingly advance the fundamental scientific understanding of the SCL mechanism in ASSLIBs and therefore boost the development of energy storage devices.

## Results

### In situ charge-density-distribution characterization of the LCO/LPSCl interface.
To visualize the SCL effect on interfacial lithium-ion transport in ASSLIBs, we first carry out in situ DPC-STEM measurements to observe the charge-density distribution at the LCO/LPSCl interface. The configuration of the in situ solid-state battery is shown in Fig. 1a–c and Supplementary Figs. 3–5. In DPC-STEM measurements, we initially obtain the electric field map caused by the SCL, the mean inner potential (MIP) difference between the cathode and electrolyte, and the possible dynamical diffraction effect (DDE). To eliminate the extraneous interferences of the MIP and DDE parts on the interface charge density, the data processing process described in detail in the "DPC characterization" subsection in the "Methods" section is applied. The obtained in situ net electric field and corresponding charge-density distribution under different bias voltages at the LCO/LPSCl interface are shown in Supplementary Fig. 6 and Fig. 1d–i as well as in Supplementary Fig. 7.

When the LCO cathode and LPSCl SE come into contact, the lithium-ion concentration on the LPSCl side of the interface will decrease to match the electrochemical potential of lithium ions between the two contacting species and make the interface reach equilibrium. Therefore, a lithium-ion-deficient negative-charge-density region will obviously be found on the LPSCl side of the interface (Supplementary Fig. 7a). Although previous reports have speculated that the SCL on the LCO side of the interface should vanish because the electronic conduction can deal with the concentration gradient of lithium ions[25,32], the unexpected finding in our study is that there still exists a lithium-ion-enriched positive-charge-density region on the LCO side of the interface (Supplementary Fig. 7a). This probably occurs because the electronic conductivity of LCO is not high ($\sim 10^{-4}\,S\,cm^{-1}$) enough to completely balance the positive net charge region[33]. In particular, LCO will exhibit an insulated state when the lithium content is higher than 0.95[34]. Consequently, an SCL with separation of lithium-ion-depleted (electrolyte side) and lithium-ion-enriched (cathode side) regions on opposite sides of the interface is formed at the LCO/LPSCl interface. However, it

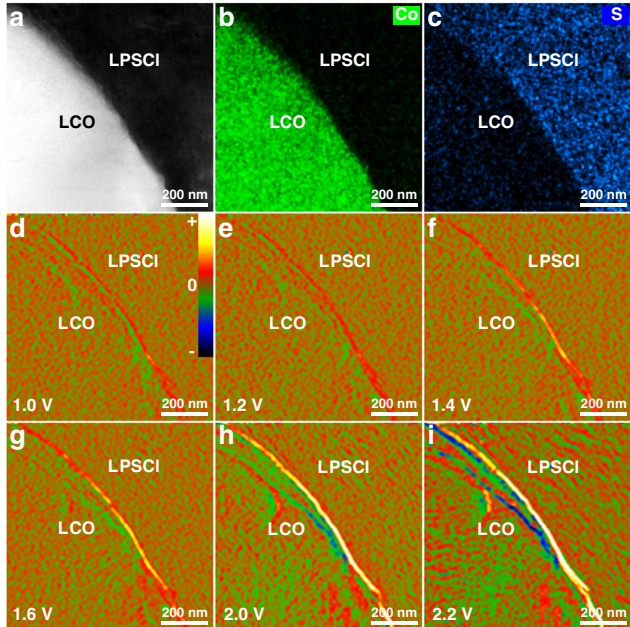

**Fig. 1 In situ charge-density-distribution characterization of the LCO/ LPSCl interface. a–c** HAADF-STEM image of the LCO/LPSCl interface **a**, and corresponding mappings of Co **b** and S **c** elements. **d–i** In situ DPC-STEM observations of net-charge-density accumulation at the LCO/LPSCl interface with bias voltages of 1.0 V **d**, 1.2 V **e**, 1.4 V **f**, 1.6 V **g**, 2.0 V **h**, and 2.2 V **i**. The color bar in Fig. 1d is defined as the relative magnitude of the positive/negative charge density.

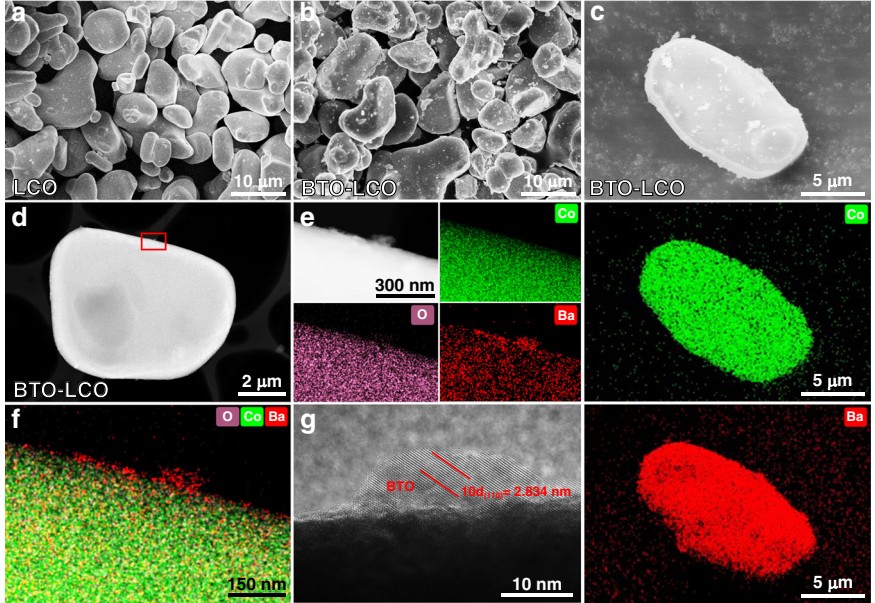

**Fig. 2 Surface structure and composition of LCO and 5 at.% BTO–LCO. a, b** Typical SEM images of LCO and BTO–LCO powders. **c** Typical SEM images and their EDS mappings of Co and Ba elements for a single BTO–LCO particle. **d** Typical HAADF-STEM image of a single BTO–LCO particle. **e, f** HAADF-STEM image and corresponding EDS mappings of Co, O, and Ba elements for the selected regions in **d**. **g** High-resolution HAADF-STEM image of the BTO–LCO particle surface.

should be noted that thus far, there have been no feasible methods to remove the electric field component due to the MIP difference from the DPC-STEM images at 0 V because the electric field due to the MIP difference at the interface changes after cathode/electrolyte contact[20,35–37]. Therefore, there is no doubt that the electric field due to the MIP difference interferes with the DPC-STEM result (only from the SCL) at 0 V to some extent.

The Li element mapping obtained by EELS can reflect the migration of Li ions after cathode/electrolyte contact[21,38,39], so we further analyze the Li and Co elemental profiles from the EELS line scan (Supplementary Fig. 8) under 0 V at the interface. On the cathode side, it can be found that interfacial lithium ions are more abundant than bulk lithium ions without BTO coating (Supplementary Fig. 8a, c). This indicates that there appears to be obvious lithium-ion enrichment on the LCO side at the interface due to lithium-ion diffusion from the electrolyte to the cathode. When combined with the DPC-STEM results (the charge-density distribution with positive and negative charges separation) of the LCO/LPSCl interface, it can be inferred that at the interface without BTO coating, the lithium-ion enrichment on the LCO side should be the main source of the positive-charge-density distribution, while the corresponding lithium vacancies should be the main source of the negative-charge-density distribution. This is because many more charges accumulate at the interface, which can neutralize the false image effect from the difference in the MIP. Therefore, the DPC-STEM result of the LCO/LPSCl interface still shows a charge-density distribution with positive and negative charges separation.

As shown in Fig. 1d–i and Supplementary Fig. 7b, the slightly negative net-charge-density region at 1.0 V on the LCO side of the interface indicates decreased positive charge accumulation when a small number of lithium ions are extracted from LCO crystal lattices and then enter the interstitial voids of the LPSCl SE (Fig. 1d). Owing to the resistances of lithium-ion-deficient layers arising from the SCL, some of the extracted lithium ions can migrate to the anode side to generate a current, while the rest are detained on the LPSCl side of the interface. The positive charge accumulation stemming from the detained lithium ions leads to

the formation of a slightly positive net-charge-density region at 1.0 V on the LPSCl side of the interface (Fig. 1d). With increasing bias voltage, more lithium ions will be extracted from the LCO crystal lattice, causing more obvious negative charge accumulation on the LCO side of the interface (Fig. 1e–i and Supplementary Fig. 7b). On the other hand, the positive charge accumulation on the LPSCl side of the interface will also be more pronounced because more lithium ions are stranded. Our in situ visualization findings first show direct experimental evidence of the resistance effect from the SCL on interfacial lithium-ion transport in ASSLIBs, which makes it possible to deeply understand the interface improvement mechanism of the built-in electric field and chemical potential coupling strategy for suppressing the SCL.

**Construction and evaluation of the built-in electric field and chemical potential coupling strategy for suppressing the SCL.** Yada et al.[40] reported that dielectric $BaTiO_3$ (BTO) nanoparticles can reduce the $LiCr_{0.05}Ni_{0.45}Mn_{1.5}O_{4-\delta}$/LiPON interface impedance stemming from the SCL by forming a reverse built-in electric field under the external electric field of the SCL. However, the detailed mechanism and universality of dielectric modification of interfacial lithium-ion transport in ASSLIBs have not been demonstrated due to the very little evidence. Consequently, discontinuous BTO nanoparticles are selected as the coating on the LCO/LPSCl interface to demonstrate the improvement mechanism of the built-in electric field and chemical potential coupling strategy for suppressing the SCL and boosting interfacial lithium-ion transport. Discontinuous BTO nanoparticle-coated LCO (BTO–LCO) cathode materials are prepared via the sol–gel method. XRD results show that crystalline BTO can be obtained at contents of 2 and 5 at.% without impurity phase $BaCO_3$, which appears in the 8 at.% BTO-coated LCO sample (Supplementary Fig. 9). Then, the surface structure and composition of the LCO and 5 at.% BTO–LCO powders at the microscale are characterized. Typical SEM images show that both the LCO and BTO–LCO powders are composed of 5−15 μm particles (Fig. 2a, b). The obvious difference is that the LCO powders display smoother

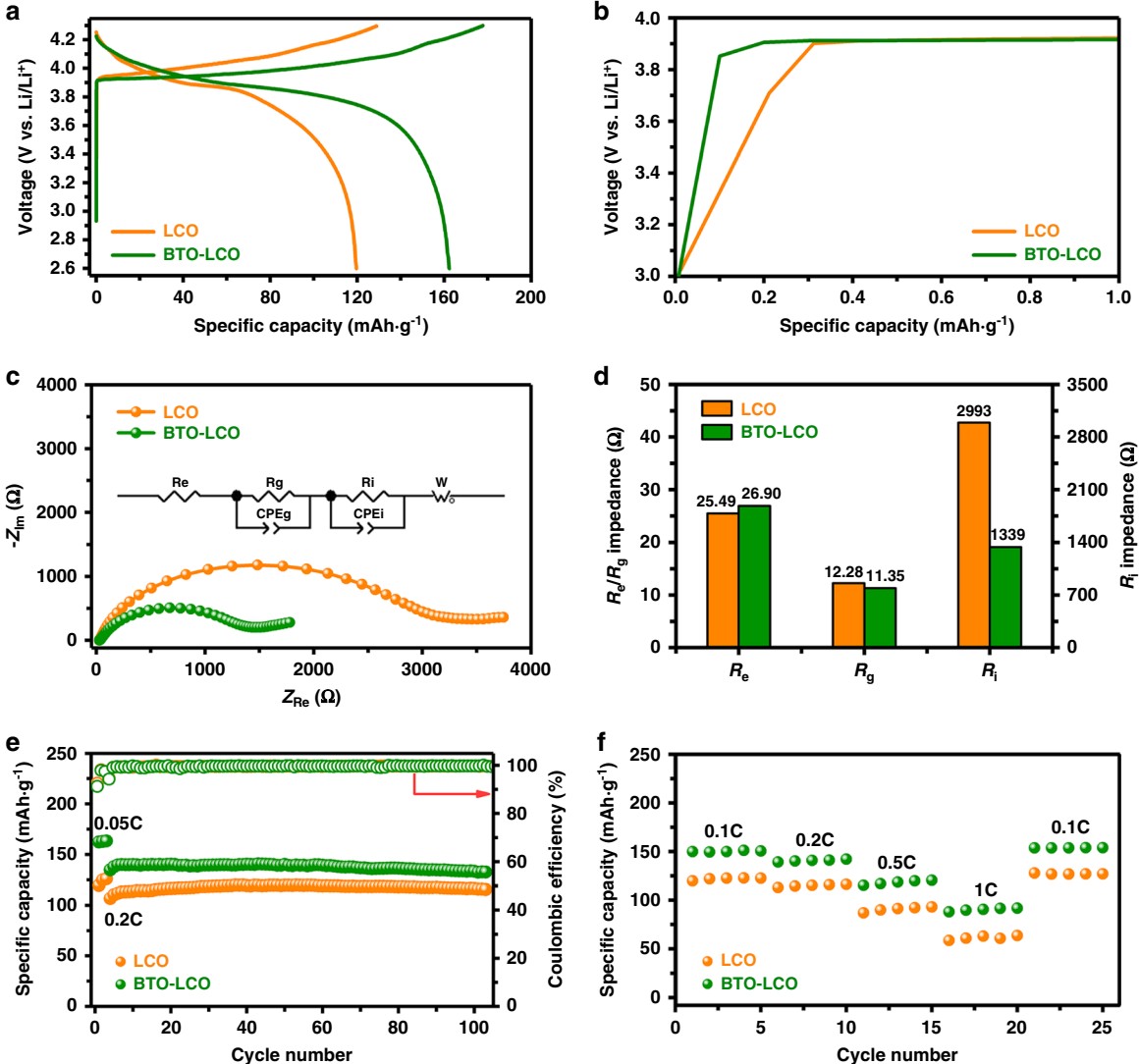

**Fig. 3 Electrochemical performances of LCO and BTO–LCO. a** First galvanostatic charge–discharge curves of LCO/In–Li and BTO–LCO/In–Li all-solid-state cells at 0.05 C. **b** Enlarged curves for the initial charge state in **a**. **c**, **d** EIS results of LCO/In–Li and BTO–LCO/In–Li all-solid-state cells after the first cycle. The inset shows the equivalent circuit used for fitting the experimental EIS data. **e** Cycling performance of LCO/In–Li and BTO–LCO/In–Li all-solid-state cells at 0.2 C after pre-cycling of three cycles at 0.05 C. **f** Rate capability of LCO/In–Li and BTO–LCO/In–Li all-solid-state cells. All cells are measured at 30 °C between 2.0 and 3.7 V (vs. In/InLi), corresponding to ~2.6–4.3 V (vs. Li/Li+).

surfaces, while there are several aggregated nanoparticles on the surface of the BTO–LCO powders. The Co and Ba energy-dispersive X-ray spectroscopy (EDS) mappings of a single BTO–LCO particle show that the Ba element is uniformly dispersed on the surface of LCO after coating (Fig. 2c), demonstrating that the aggregated nanoparticles are BTO coatings. To further investigate the nanoscale distribution of BTO, high-angle annular-dark-field scanning transmission electron microscopy (HAADF-STEM) is carried out for BTO–LCO powders (Fig. 2d–g). The HAADF-STEM images of a single BTO–LCO particle show discontinuously distributed BTO nanoparticles on the surface of LCO, which is proven by the mapping results of Co, O, and Ba elements and the (110) lattice stripes of BTO. Consequently, it can be concluded that we have successfully coated the LCO powders with discontinuously distributed BTO nanoparticles.

To study the effects of the BTO nanoparticles on interfacial lithium-ion transport, the electrochemical performances of LCO/In–Li and BTO–LCO/In–Li all-solid-state cells are tested. As shown in Fig. 3a, the LCO only exhibits a discharge capacity of

119.6 mAh g$^{-1}$ even at the low current density of 0.05 C in the initial cycle. By contrast, the BTO–LCO displays a relatively high initial discharge capacity (162.3 mAh g$^{-1}$) at the same current density. Additionally, it can also be found that the BTO–LCO/In–Li all-solid-state cell exhibits a smaller polarization than the LCO/In–Li all-solid-state cell (Supplementary Fig. 10). From the enlarged curves for the initial charge state (Fig. 3b) of Fig. 3a, BTO–LCO shows a shortened potential slope compared with LCO. Takada et al.[25,41] has noted that the potential slope at the beginning of the charge curve prior to the 4 V potential plateau mainly stems from the SCL influence, which blocks lithium-ion conduction and enhances the interfacial impedances. Therefore, it can be inferred that the coated BTO nanoparticles effectively suppress the SCL formation and decrease the interfacial resistances. As shown in Fig. 3c, d, the electrochemical impedance spectroscopy (EIS) results of the LCO/In–Li and BTO–LCO/In–Li all-solid-state cells after the first cycle reveal that their electrolyte resistance ($R_e$) and grain boundary resistance ($R_g$) are consistent ($\approx$26 Ω for $R_e$; $\approx$12 Ω for $R_g$). By sharp contrast, the interfacial resistance ($R_i$) of BTO–LCO/In–Li ($\approx$1339 Ω) is much

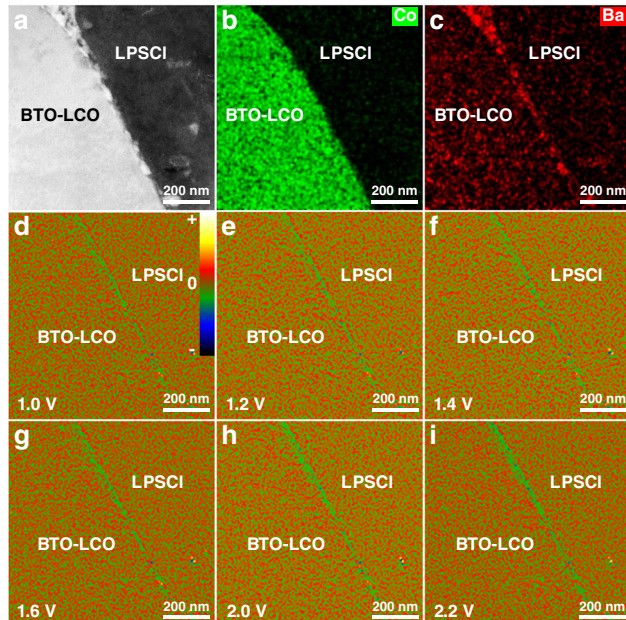

**Fig. 4 In situ charge-density-distribution characterization of the BTO–LCO/LPSCl interface. a–c** HAADF-STEM image of the BTO–LCO/LPSCl interface **a**, and corresponding mappings of Co **b** and Ba **c** elements. **d–i** In situ DPC-STEM observations of net-charge-density accumulation at the BTO–LCO/LPSCl interface with bias voltages of 1.0 V **d**, 1.2 V **e**, 1.4 V **f**, 1.6 V **g**, 2.0 V **h**, and 2.2 V **i**. The color bar in **d** is defined as the relative magnitude of the positive/negative charge density.

lower than that of LCO/In–Li (≈2993 Ω), demonstrating that the BTO nanoparticles effectively restrain the SCL formation.

Moreover, the stable cycling performance and higher capacity of BTO–LCO suggest a stable chemical/electrochemical interface between BTO–LCO and the sulfide electrolyte as well as the sustained and steady modification effect of BTO on the interfacial lithium-ion transport dynamics (Fig. 3e). On the other hand, BTO–LCO also demonstrates a better rate capability than LCO due to the suppressed SCL effect. As shown in Fig. 3f, BTO–LCO can show a specific capacity of up to 92 mAh g$^{-1}$ even at the high current density of 1 C in ASSLIBs, whereas the discharge capacity of LCO only reaches 60 mAh g$^{-1}$ at 1 C. Therefore, the discontinuously distributed BTO nanoparticles significantly enhance the capacity and rate capability of LCO-based ASSLIBs by reducing the interfacial resistances resulting from the SCL.

**In-situ charge-density-distribution characterization of the BTO–LCO/LPSCl interface.** The configuration of the BTO–LCO/LPSCl interface is shown in Fig. 4a–c. It can be found that BTO nanoparticles discontinuously distribute on the LCO/LPSCl interface (Fig. 4c). When ferroelectric BTO nanoparticles are coated on the LCO interface, BTO will generate permanent reverse electric dipoles due to the spontaneous polarization under the electric field effects of the formed SCL. The electric dipoles in the BTO should be arranged such that the SCL is reduced: their negative poles should face the positive charges on the LCO cathode side, while their positive poles should face the other direction[40]. Under the built-in electric field of BTO, the lithium ions will redistribute at the LCO/LPSCl/BTO triple-phase interface (TPI). Driven by the Coulomb interaction, both lithium ions in LPSCl initially located behind the BTO (near the positive pole side of BTO) and lithium ions in LCO originally located across from the BTO (near the negative pole side of BTO) will migrate towards the vicinity of the LCO/LPSCl/BTO TPI to maintain

local charge neutralities. Owing to the limited action scope of the Coulomb interaction, the number of lithium ions that migrate to the interface decreases when moving away from the LCO/LPSCl/BTO TPI. Therefore, overall, the lithium-ion-deficient negative-charge-density region on the LPSCl side and the lithium-ion-enriched positive-charge-density region on the LCO side should be significantly restrained. Unfortunately, the result of the corresponding charge-density distribution at 0 V was not obtained by DPC-STEM due to the greater impact from the MIP difference (Supplementary Fig. 11a). This is because much less charge accumulates at the interface after BTO modification. Accordingly, the false image effect from the MIP difference is particularly obvious. Therefore, the DPC-STEM result of the BTO–LCO/LPSCl interface does not show an obvious charge-density distribution with positive and negative charges separation but shows a false image of only the positive-charge-density layer (Supplementary Fig. 11a). Nevertheless, it can still reveal the charge-density distribution of the BTO–LCO/LPSCl interface from this side. To elaborate the lithium-ion distribution under 0 V at the interface after BTO coating, we also analyze the corresponding Li and Co elemental profiles from the EELS line scan (Supplementary Fig. 8b, d). It can be found that the lithium-ion enrichment on the LCO side is clearly suppressed after BTO coating (Supplementary Fig. 8b, d). Such a rearrangement of the lithium-ion distribution will lead to fast continuous pathways of lithium-ion conduction, thus significantly improving the interfacial migration kinetics. When adding a bias voltage to the BTO–LCO/LPSCl interface, the result is obtained by subtracting the corresponding electric field result at 0 V before the partial differential treatment. It can be found that there is no obvious positive net-charge-density region (i.e., positive charge accumulation) at the interface (Fig. 4d–i and Supplementary Fig. 11b) and that a higher current is generated (Supplementary Fig. 12). Therefore, the larger battery current and lack of positive charge accumulation at the BTO–LCO/LPSCl interface successfully demonstrate the significantly accelerated interfacial lithium-ion transport. These results also prove the existence of fast continuous interfacial lithium-ion conduction pathways resulting from the effects of ferroelectric BTO.

## Discussion

The SCL is usually formed at the cathode/electrolyte interface, which brings about separation of lithium-depleted (electrolyte side) and lithium-enriched (cathode side) regions on opposite sides of the interface (Fig. 5a)[21,25,42–44]. In particular, the lithium-depleted layer on the SE side is highly resistive due to the lack of charge carriers and becomes a bottleneck of lithium-ion transport. According to conventional wisdom, interposed cathode interface buffering layers (CIBLs) possessing high lithium-ion conduction and strong lithium-ion attraction can restrain the SCL formation to some extent[25]. Typical material systems meeting these prerequisites are oxide-based fast Li-ion conductors, such as LiNbO$_3$, Li$_3$BO$_3$–Li$_2$CO$_3$[18,45], LiNb$_{0.5}$Ta$_{0.5}$O$_3$[46,47], and Li$_{0.35}$La$_{0.5}$Sr$_{0.05}$TiO$_3$[48]. Nevertheless, it should be noted that the CIBLs form two kinds of double-phase interfaces between the cathode and electrolyte (Fig. 5b), which will create an additional interfacial diffusion barrier[44]. Therefore, fast Li-ion conductor coatings are not the best strategy for solving the SCL issue.

The SCL formation is an interfacial charge redistribution process driven by the chemical potential difference between the cathode and electrolyte, which should also be influenced by an electric field. Thus, introducing a built-in electric field at the cathode/electrolyte interface will suppress the SCL formation and even create lithium-ion conduction pathways by adjusting the interface charge redistribution at the TPI without adding an

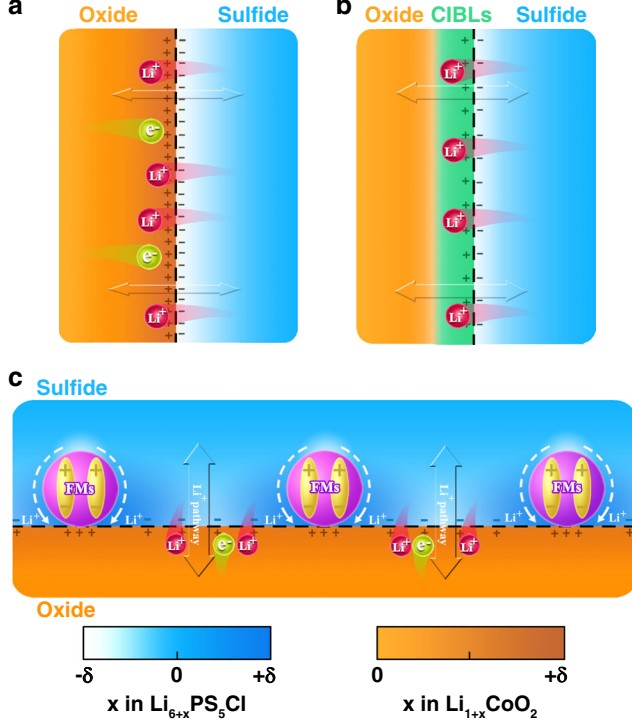

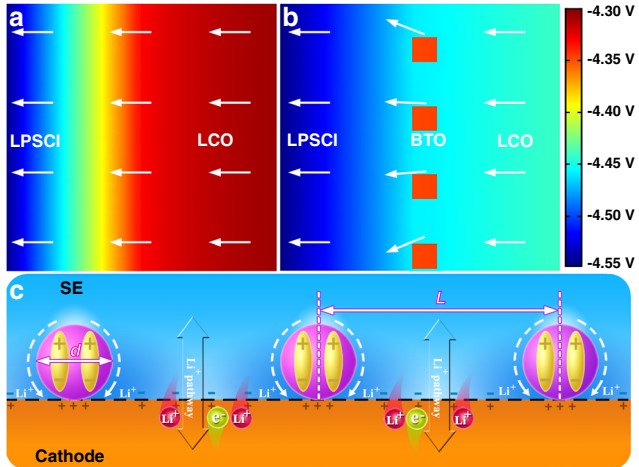

**Fig. 5 Schematic illustration of interface charge redistribution between oxide cathodes and sulfide SEs after different interface engineering approaches. a** Initial interface. **b** Double-phase interface with fast Li-ion conductor CIBLs. **c** TPI with discontinuous ferroelectric nanoparticles. The regions with light colors correspond to the relative Li-ion deficiency. FMs ferroelectric materials.

**Fig. 6 FEM simulations of the effects of BTO nanocrystals. a, b** Simulation results of the internal electrical field for the LCO/LPSCl and BTO–LCO/LPSCl interfaces. The arrows represent the direction of the internal electrical field. **c** Schematic illustration of broader interface engineering for suppressing the SCL. The $L$ in **c** is defined as the distance between neighboring BTO nanoparticles.

additional interfacial diffusion barrier[40,49–54]. Dielectric materials have been demonstrated to build an internal electric field under an external electric field, stress, or temperature. Taking a discontinuous nano-ferroelectric material (FM) as the model, our electrochemical and in situ DPC-STEM results discussed above have thoroughly demonstrated the feasibility of the built-in electric field and chemical potential coupling strategy for

suppressing the SCL and boosting interfacial lithium-ion transport (Fig. 5c). Furthermore, our theoretical calculations further confirm that discontinuous nano-FMs can feasibly suppress the SCL in ASSLIBs. The internal electric field mappings at the LCO/LPSCl and BTO–LCO/LPSCl interfaces are calculated through FEM simulations based on the semiconductor analogy (Supplementary Fig. 13)[44,50,55,56]. As shown in Fig. 6a, the orientation of the interface internal electrical field is from the LCO cathode to the LPSCl electrolyte without an applied voltage, which suggests that there is an SCL at the LCO/LPSCl interface. After coating with discontinuous BTO nanoparticles, the gradient of the internal electrical field is greatly reduced, as expected (Fig. 6b), demonstrating that the SCL is suppressed.

More broadly, to drive lithium-ion redistribution at the TPI and thus form lithium conduction pathways between neighboring coated nanoparticles (CNs), the built-in electric field intensity $E_{in}$ of the CNs should be strong enough (Fig. 6c). According to the Lorentz hypothesis, the built-in electric field intensity of CNs ($E_{in}$) is directly proportional to the electric field intensity $E_{SCL}$ of the SCL and the dielectric constant of the CNs ($\varepsilon_{CNs}$). The SCL is caused by the chemical potential difference between the electrolyte and cathode ($\phi_{electrolyte}-\phi_{cathode}$), and the dielectric constants of CNs strongly depend on their particle sizes ($d_{CNs}$)[57,58]. Furthermore, the redistribution of lithium ions at the TPI is also determined by $E_{in}$ and the distance $L$ between neighboring BTO nanoparticles. Thus, the particle size and distribution of CNs influence the SCL formation and fast lithium-ion conduction pathways, leading to different interface resistances and reversible capacities. As demonstrated by the electrochemical results shown in Supplementary Figs. 14 and 15, the enlarged curves for the initial charge state of LCO and 2, 5, and 8 at.% BTO-coated LCO cathodes show that the inhibiting effect of BTO on the SCL increases from 2 to 5 at.% BTO content, while it sharply decreases at 8 at.%. Their corresponding polarizations and reversible capacities have the same change trends. Then, it can be concluded that the built-in electric field and chemical potential coupling effect can be optimized by adjusting the built-in electric field intensity $E_{in}$-related factors, such as the interfacial chemical potential difference, dielectric constants (kinds and sizes of dielectric materials), and distance between neighboring CNs.

In summary, we have achieved in situ visualization of the SCL effect on interfacial lithium-ion transport and showed direct experimental evidence of interfacial lithium-ion accumulation in ASSLIBs using in situ DPC-STEM. Furthermore, we demonstrated in detail an innovative built-in electric field and chemical potential coupling strategy to reduce the SCL effect in sulfide-based ASSLIBs. The in situ DPC-STEM and FEM simulation results confirm that the discontinuously coated BTO nanoparticles can effectively suppress the SCL formation and lead to fast continuous interfacial lithium-ion conduction pathways, thus significantly improving the interfacial migration kinetics between cathode materials and the sulfide electrolyte. Therefore, the BTO–LCO/In–Li all-solid-state cells can exhibit a high discharge capacity of nearly 140 mAh g$^{-1}$ at 0.2 C when the upper cutoff voltage is set to 4.3 V. Excitingly, the BTO–LCO cathode can still display a specific capacity of up to 92 mAh g$^{-1}$ at 1 C, which is much higher than that (60 mAh g$^{-1}$) of the LCO cathode at the same current density in ASSLIBs. Our findings will strikingly advance the fundamental scientific understanding of the SCL mechanism in ASSLIBs and therefore open up a new direction for interface engineering of energy storage devices.

## Methods
**Material synthesis**. BTO buffering layers were coated on LCO particles via the sol-gel method. The BTO coating contents were set to 2, 5, and 8 at.% LCO. Typically, commercial high-voltage LCO powders (MGL New Materials) were first

ultrasonically dispersed in absolute ethanol (AR, Sinopharm Chemical Reagent). Then, barium acetate (99.99%, Aladdin) was dissolved in acetic acid (1.0 M, Aladdin), and titanium butoxide (99+%, Alfa Aesar) was dissolved in 2-methoxyethanol (99.8%, Aladdin). After that, these solutions were slowly added to the LCO dispersion simultaneously under vigorous stirring at 70 °C. After continuous stirring for 12 h, the obtained gel was dried and annealed at 700 °C for 20 h to obtain BTO–LCO powders. The 5 at.% BTO-coated LCO powders were selected as the coated samples for in-depth analysis because they exhibited the best cycling performance among the three coated samples based on our electrochemical measurements (Supplementary Fig. 15). The LPSCl SE powders were prepared by ball milling a stoichiometric mixture of Li₂S (99.9%, Alfa Aesar), P₂S₅ (≥99.5%, Macklin), and LiCl (99%, Alfa Aesar) at 600 rpm for 10 h with ZrO₂ balls. After that, the ball-milled powders were heat-treated at 550 °C for 5 h in an Ar atmosphere. The total conductivity of the synthesized LPSCl SE (Supplementary Fig. 16) using blocking stainless steel electrodes was $3.2 \times 10^{-3}$ S cm⁻¹ (Supplementary Fig. 17, Supplementary Note 1).

**Preparation and transfer of the in situ solid-state battery.** The sulfide solid-state electrolyte is extremely sensitive to moisture, so we protected the samples from direct air exposure at every experimental stage. After the bulk solid-state battery without an anode was prepared in an Ar-filled dry glovebox, the solid-state battery pellet was sealed in the Ar-filled chamber of a special sample stage in the glovebox. Then, the stage was removed from the glovebox and transferred to an FIB-SEM system (FEI Helios Nanolab 460HP). A relatively flat area on the solid-state battery pellet in the microscopic field of view was selected as the further processing area. First, a thin Pt protective layer was deposited on the selected area under an electron beam with relatively low energy. After that, the selected specimen (10 × 1.5 × 1 μm) was taken and placed across the middle (observation area) of a chip. Then, the specimen was fixed and connected to a Pt electrode at both ends by redepositing a thin Pt layer. Finally, the specimen in the observation area was thinned by a focusing ion beam (FIB). The fabricated in situ solid-state battery chip was sealed in the chamber of the special sample stage (the chamber was at vacuum at this moment). Then, the stage was removed from the FIB-SEM system and transferred into another Ar-filled dry glovebox. In the glovebox, the chip was removed from the stage and sealed in the Ar-filled chamber of another special sample stage of an ALD system. After ALD coating of Al₂O₃ onto the chip, the chip was placed on a DENS single tilt specimen holder to carry out the in situ DPC-STEM experiments.

**Material characterization.** XRD measurements were conducted using an X-ray diffractometer (Rigaku SmartLab) equipped with a Cu Kα radiation source ($\lambda_1$ = 1.54060 Å, $\lambda_2$ = 1.54439 Å). To avoid exposure to air, the LPSCl powders were sealed in the sample holder with polyimide film. SEM measurements were carried out using a field emission SEM (Hitachi SU-8010) equipped with an energy-dispersive spectrometer (EDS, Oxford X-Max 80). HAADF-STEM images and mappings were obtained by a JEOL ARM200F (JEOL, Tokyo, Japan) STEM with an accelerating voltage of 200 kV with a thermal field-emission gun and a probe Cs corrector (CEOS GmbH, Heidelberg, Germany). For in situ solid-state battery experiments, the bias voltage range was 0−2.2 V in the electron microscope. EDS-mapping images were obtained by using an FEI Talos F200X with a field-emission gun and an accelerating voltage of 200 kV. X-ray absorption spectra at the P K-edge and the S K-edge were measured using fluorescence yield mode (~200 nm depth) at the 16A beamline (1800−7500 eV) at the Taiwan Synchrotron Radiation Research Center.

**Electrochemical measurements.** The LCO/In–Li ASSLIBs were tested by using STC-SB polyaryletheretherketone (PEEK) mold cells with a 10 mm diameter (Hefei Kejing Mater. Tech. Co. Ltd., China) at 30 °C. An 80 mg amount of the SE powder was first added into the cylinder, followed by uniaxial pressing at 150 MPa for 2 min. The composite cathode was prepared by hand mixing LCO and LPSCl in the mass ratio of 70:30 in an agate mortar for 30 min. A 10 mg amount of this mixed powder was homogeneously distributed on one side of the preformed SE pellet, followed by uniaxial pressing at 370 MPa for 2 min. After the second pressing step, a thin indium foil (0.1 mm) with a 9 mm diameter and a thin lithium foil (0.25 mm) with a 3 mm diameter were successively added to the other side of the SE pellet and pressed at 150 MPa for 2 min. After that, a constant pressure was applied to the cell using the screw of the stainless steel framework, which was kept constant during the electrochemical tests. The mass loading of the LCO cathode material was 8.9 mg cm⁻². Galvanostatic cycling tests of the cells were conducted using a Land battery test system (Land CT2001A, Wuhan Land Electronic Co. Ltd., China) in the voltage range from 2.0 to 3.7 V (vs. In/InLi), which corresponds to ~2.6–4.3 V (vs. Li/Li⁺). The C-rate of 1 C corresponds to 140 mA g⁻¹. The EIS tests were performed over a frequency range of 1–5 mHz with an applied amplitude of 10 mV by an electrochemical working station (Biologic VMP-300).

**DPC characterization.** DPC results were obtained using an FEI Talos F200X with a field-emission gun and an accelerating voltage of 200 kV. According to the established DPC methodology[23,24,27−29], the electric field (**E**) at a site with the coordinates of (x, y) in a sample can be obtained from four DPC images (see a set

of typical DPC images in Supplementary Fig. 5) using the following equation:

$$\mathbf{E} = -\frac{R^2 - r^2}{RC} \cdot \frac{m_{rel} v_{rel}^2}{e} \cdot \frac{1}{t} \cdot \frac{I_{A-C}\hat{x} + I_{B-D}\hat{y}}{I_{sum}}, \tag{1}$$

where $R$ is the radius of the electron diffraction disk, $r$ is the inner (hole) radius of the annular DPC detector, $C$ is the camera length, $m_{rel}$ and $v_{rel}$ are the relativistic electron mass and velocity, respectively, $t$ is the absolute thickness of the sample, which can be obtained by EELS[54] (Supplementary Fig. 18, Supplementary Note 2), $I_{A-C}$ is the difference between the electron intensities at the site in the DPC images obtained by Segments A and C of the DPC detector, $I_{B-D}$ is the difference between those obtained by Segments B and D, $I_{sum}$ is the sum of the intensities at the site in the DPC images obtained by the four segments, and $\hat{x}$ and $\hat{y}$ denote the x and y axis directions. Using the equation and DPC images, the electric field distribution of a sample can be attained. It should be noted that the electric field obtained from DPC without a bias voltage is determined by the SCL, the MIP difference between the cathode and electrolyte, and the possible DDE, named .., $\mathbf{E}_j^{MIP}$ and $\mathbf{E}_j^{DDE}$ ($j = x$ or $y$), respectively. That is,

$$\mathbf{E}_j^{0V} = \mathbf{E}_j^{SCL(0V)} + \mathbf{E}_j^{MIP} + \mathbf{E}_j^{DDE}. \tag{2}$$

When a bias voltage is applied to the battery, the electric field $\mathbf{E}_j^{bias}$ should change to

$$\mathbf{E}_j^{bias} = \mathbf{E}_j^{SCL(bias)} + \mathbf{E}_j^{MIP} + \mathbf{E}_j^{DDE}, \tag{3}$$

where $\mathbf{E}_j^{SCL(bias)}$ is the SCL-caused electric field at the bias voltage. To eliminate the extraneous interferences of the electric fields due to the MIP difference and possible DDE, the electric field result at 0 V is subtracted from that at the bias voltage. This goal of removing the electric field (MIP difference) effect at a bias voltage can be easily achieved because the formed electric field due to the MIP difference after cathode/electrolyte contact is constant[20,35–37]. That is,

$$\Delta\mathbf{E}_j^{net\,(bias)} = \mathbf{E}_j^{bias} - \mathbf{E}_j^{0V} = \mathbf{E}_j^{SCL(bias)} - \mathbf{E}_j^{SCL(0V)}, \tag{4}$$

According to Gauss's law, the charge-density map can be obtained by calculating the differential of the electric field map[23,24,59–62]. Then, according to the relation between the electric field (**E**) and charge density ($\rho$)[49,51]

$$\rho = \frac{\varepsilon_0}{e}\left(\frac{\partial E_x}{\partial x} + \frac{\partial E_y}{\partial y}\right) \tag{5}$$

the charge density can be obtained by calculating the divergence of the electric field strength, where $\varepsilon_0$ is the vacuum dielectric constant. We used the plugin of Holowork in DigitalMicrograph® to obtain $\frac{dE_x}{dx}$ and $\frac{dE_y}{dy}$. The charge density can then be acquired by adding $\frac{dE_x}{dx}$ and $\frac{dE_y}{dy}$.

**FEM simulations.** The built-in electric field model is implemented through FEM simulations based on the commercial software COMSOL Multiphysics. For simplicity, a two-dimensional FEM model is implemented. As an analogy to the semiconductor module, a predefined semiconductor interface is applied. The drift-diffusion and Poisson equations are solved by the FEM. A set of coupled partial differential equations is solved for the electric potential and the electron and lithium-ion concentrations. The corresponding boundary conditions on the electrode side and the electrolyte side are specified as 0 V. The internal electric field is caused by the migration of lithium ions. The material parameters are consistent with the experimental parameters. The active material is modeled as a rectangular plate with a length of 500 nm and a width of 400 nm. The electrolyte is modeled as a rectangular plate perfectly bonded to the active material with a length of 500 nm and a width of 100 nm. BTO is simulated as squares with a side length of 20 nm near the cathode–electrolyte interface. The spacing of each BTO nanoparticle is 50 nm.

## Data availability

The data that support the findings of this study are available from the corresponding author upon reasonable request.

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

## Acknowledgements

This work was supported by the National Key R&D Program of China (2018YFB0104300), the National Natural Science Foundation of China (21975274, U1706229, 11604241, 51971157, 21603161, 61705115, 11902144, 51971157, and 51761165012), the Young Elite Scientists Sponsorship Program by Tianjin, the Strategic Priority Research Program of the Chinese Academy of Sciences (XDA22010600), the National Natural Science Foundation for Distinguished Young Scholars of China (51625204), the Youth Innovation Promotion Association of CAS (2016193), the Key Research and Development Plan of Shandong Province, China (2018GGX104016), the Tianjin Science Fund for Distinguished Young Scholars (19JCJQJC61800), the Tianjin Municipal Science and Technology Commission (19JCQNJC15100), the National Program for Thousand Young Talents of China, DICP & QIBEBT Fund (Grant No. DICP and QIBEBT UN201707) and QIBEBT (ZZBS201808). The authors gratefully acknowledge Prof. Shuping Pang (Qingdao Institute of Bioenergy and Bioprocess Technology, Chinese Academy of Sciences, Qingdao 266101, China), Prof. Lin Gu and Dr. Qinghua Zhang (Institute of Physics, Chinese Academy of Sciences, Beijing 100190, China) for the valuable discussions and kind help in revising the manuscript.

## Author contributions

L.L.W., J.M., and G.L.C. conceived the idea. L.L.W., X.R.Y., and X.W.S. synthesized the materials and performed XRD, SEM, XAS, and electrochemistry measurements. R.C.X., C.L., and J.L. performed the in situ DPC-STEM measurements and analysis. B.B.C. and C.J.X. performed the FEM simulations and analysis. Z.W.H. and T.-S.C. helped to analyze the XAS data. L.L.W. and J.M. wrote the paper with critical input from S.M.D. and L.Q.C. All authors edited and approved the manuscript.

## Competing interests

The authors declare no competing interests.
