## [Peer Review File · Nature Communications]

Reviewers' Comments:

Reviewer #1:

Remarks to the Author:

The authors reported in-situ study of SCL formed at LCO/LPSCI interface by DPC STEM. This is indeed interesting new challenge and should give very important information if obtained data were properly interpreted. However, the authors interpretation of DPC STEM results appears to be still missing some points. To visualize SCL at heterointerfaces, it is essential to differentiate the pure SCL contribution from the positive and negative charge layers resulted from the mean-inner-potential difference across the two different phases forming the heterointerfaces. However, the authors do not discuss the mean-inner-potential effect. Indeed, the mean-inner-potential effect can also accounts for the negative and positive charge visualized in Fig. 1d. The LCO/LPSCI/BTO triple-phase interface is more complex structure, so the mean-inner-potential effect in the DPC image should be discussed and differentiated from the pure SCL effect. Furthermore, polarities of the BTO nanoparticles may also affect the DPC images, but not considered here. Therefore, I cannot recommend this paper for Nature Communications.

Minor comments:

On Lines 178-187, the description of the phenomena is not very clear. The authors should explain more clearly and concretely how lithium ions migrate and the charge distribution varies.

In Fig.4d, only the positive charge layer is discussed. How does the total charge neutrality maintain?

More details of FEM simulations should be described. What chemical potentials of lithium ion were assumed?

In Fig. 6b the total potential difference looks smaller than Fig. 6a. The lithium diffusion may stop when the chemical potential and electrostatic potential is balanced. Why does the lithium diffusion stop in the case of Fig. 6b?

In Fig. 4e-i, it is not clear if the charge density at 0V is subtracted similarly in Fig. 1.

Reviewer #2:

Remarks to the Author:

High interface impedance is one of the bottle neck in the solid state battery technology, and the so-called space charge layers (SCL, also called electrical double layer) might be one of the culprit. Direct microscopy imaging of the SCL effect has hitherto not possible. By using the latest imaging technology called differential phase contrast – scanning transmission electron microscopy (DPC-STEM), the authors demonstrate the possibility of visualizing the SCL effect directly. Furthermore, they show that introducing built-in electric field on the cathode/electrolyte interface, it is possible to suppress the formation of SCL, thus benefitting the interface ion transport. The report constitutes a timely new contribution to the solid state battery research field, and it advances significantly our understanding to the SCL effect on the impedance of solid state batteries. The paper is well written, and the quality is high. I thus recommend publication of the manuscript in Nature Communications after the authors address the following minor issues.

1. The sulfide solid state electrolyte is extremely sensitive to moisture. The authors did not describe in detail how they fabricate thin foil TEM samples from a bulk solid state battery. Neither do they provide details regarding how the samples are transferred from air to FIB, and then from FIB to TEM sample holder, and how the holder was inserted into the TEM without air exposure. They do show that by coating Al₂O₃ onto the sample, air damage to the sample was avoided. But how was this done and at what stage? Any air exposure for even a nanosecond renders the SE useless. The air sensitivity of sulfide electrolyte is actually what impedes the application of in-situ TEM in this important research

field.

2. Based on Fig. 2 in ref. 20, upon biasing the battery, the potential in the LCO side (essentially caused by the changes of Li ion concentration) changes with the biasing voltage accordingly, however, such changes are not reflected in the DPC imaging. Can the authors provide an explanation for this?

3. Is it possible to provide a quantitative measurement of the width of the SCL layer? And how does it vary with biasing? How high is the voltage and how strong is the field produced by the SCL?

Point-by-point responses to the reviewers' comments

Replies to Reviewer #1:

The authors reported in-situ study of SCL formed at LCO/LPSCI interface by DPC STEM. This is indeed interesting new challenge and should give very important information if obtained data were properly interpreted. However, the authors interpretation of DPC STEM results appears to be still missing some points. To visualize SCL at heterointerfaces, it is essential to differentiate the pure SCL contribution from the positive and negative charge layers resulted from the mean-inner-potential difference across the two different phases forming the heterointerfaces. However, the authors do not discuss the mean-inner-potential effect. Indeed, the mean-inner-potential effect can also accounts for the negative and positive charge visualized in Fig. 1d. The LCO/LPSCI/BTO triple-phase interface is more complex structure, so the mean-inner-potential effect in the DPC image should be discussed and differentiated from the pure SCL effect. Furthermore, polarities of the BTO nanoparticles may also affect the DPC images, but not considered here. Therefore, I cannot recommend this paper for Nature Communications.

Response: Thank you very much for your kind comments and helpful suggestions. We have revised the manuscript according to your kind suggestions. The revisions are highlighted with yellow in the manuscript and supplementary information.

(1) To visualize SCL at heterointerfaces, it is essential to differentiate the pure SCL contribution from the positive and negative charge layers resulted from the mean-inner-potential difference across the two different phases forming the heterointerfaces. However, the authors do not discuss the mean-inner-potential effect. Indeed, the mean-inner-potential effect can also accounts for the negative and positive charge visualized in Fig. 1d. The LCO/LPSCI/BTO triple-phase interface is more complex structure, so the mean-inner-potential effect in the DPC image should be discussed and differentiated from the pure SCL effect.

Response: Thank you very much for your useful suggestions. We fully agree that it is essential to differentiate the pure SCL contribution from the positive and negative charge layers resulted from the mean-inner-potential (MIP) difference across the two different phases forming the heterointerfaces. In the initial manuscript, we only focused on differentiating the pure SCL contribution from the effect of the MIP and possible dynamical diffraction at bias voltage by subtracting the corresponding electric-field result at 0 V before the partial differential treatment,¹⁻³

but neglected the relevant discussions without bias voltage. Following your suggestions, we have added the corresponding discussions on page 4 and 8 in the manuscript, and page 7, 10–12 in the supplementary information .

The MIP is a volumetric average of the electrostatic potential in a material with respect to a distant vacuum reference region at 0 V.^{4,5} The MIP is an intrinsic material property, which depends on its structure, elemental composition, and electronic configuration. The MIP originates from the opposing electric field contributions of the positive atomic nuclei in the material, and partial screening by dispersed electron clouds. Therefore, the MIP is always a net positive potential, and equal in the same material^{4,6}. Thus, the electric field caused by MIP is a constant in our DPC-STEM specimens with uniform thickness. The charge density we offered is obtained by partially differentiating the electric-field. The partial differential result of the equal electric field is zero. As a result, the effect of MIP on the charge distribution can be eliminated in the same material. Although the dynamical diffraction can affect the electric field of the sample with different crystal orientations,⁷⁻⁹ similarly, the effect of the dynamic diffraction can also be eliminated by the partial differential treatment. However, the partial differential treatment unavoidably brings about a sudden change at the interface (also called “edge effect”), which will influence the charge distribution at the interface to some extent. To explain more clearly, the electric field caused by MIP of LCO and LPSCI is assumed to be 50 V/nm and 80 V/nm, respectively (**Supplementary Fig. 10a**). After the same partial differential treatment as before, it can be found that the effect of MIP has been eliminated except obvious “edge effect” (**Supplementary Fig. 10b**). As for the LCO/LPSCI interface without bias voltage, much more charges are accumulated at the interface, which can neutralize some effect of false image from “edge effect”. Therefore, the DPC-STEM result of LCO/LPSCI interface can still appear the charge distribution with positive and negative charge separation. By comparison, much less charges are accumulated at the interface after BTO modification. Accordingly, the effect of false image from “edge effect” is particularly obvious. Thereupon, the DPC-STEM result of LCO/LPSCI interface does not appear the obvious charge distribution with positive and negative charge separation, but appears a false image of only the positive charge layer. Since there is some inaccuracy in the DPC-STEM results without bias voltage, we have moved these results to the supplementary information just for reference because it can still explain the charge distribution of

BTO-LCO/LPSCI interface from the side. The corresponding discussions have been supplemented on page 4 and 8 in the manuscript, and page 7, 10–12 in the supplementary information .

Supplementary Figure 10. Schematic illustration the “edge effect” on the DPC result of the charge distribution at the interface without bias voltage. a, The assumed electric-field results caused by MIP of LCO (e.g. 50 V/nm) and LPSCI (e.g. 80 V/nm). b, The partial differential results of Supplementary Fig. 10a.

(2) Furthermore, polarities of the BTO nanoparticles may also affect the DPC images, but not considered here.

Response: Thank you very much for your kind suggestion. The electric field from the SCL or the polarity of the BTO nanoparticles does deflect electron beam slightly and thus affect the DPC images, but this deflection effect is exactly the basis of our DPC imaging.¹⁰⁻¹² However, the DPC technique really can't differentiate the individual effect from the polarity of the BTO nanoparticles on the DPC images, because the obtained DPC image is the integrated result of the SCL and polarity of the BTO nanoparticles after modification. Nevertheless, there is no doubt that the polarity of the BTO nanoparticles does affect the distribution of SCL. What we are concerned about in this manuscript is the evolution of SCL at bias voltage after BTO modification. And the obtained DPC results do indicate that much less charges are accumulated at the interface after BTO modification, which is the effect from the polarity of the BTO nanoparticles. Moreover, the effect from the polarity of the BTO nanoparticles on the SCL has also been confirmed by FEM simulation results, which is consistent with the *in-situ* DPC results.

On the other hand, actually, the quantitative measurement of the evolution of the SCL layer and the polarity of the BTO nanoparticles during the charge and discharge process is an attractive but

challenging project. In this regard, *in-situ* HADDF-STEM and ABF-STEM characterization can be more effective, because there has been reported that the polarity of BaTiO₃ can be calculated by measuring the average Ti displacement using HADDF-STEM combined with ABF-STEM.¹³⁻¹⁵ Therefore, we will give more *in-situ* HADDF-STEM and ABF-STEM results with quantitative information in our next work.

Minor comments:

On Lines 178-187, the description of the phenomena is not very clear. The authors should explain more clearly and concretely how lithium ions migrate and the charge distribution varies.

Response: Thank you very much for your useful suggestion. To better understand how lithium ions migrate and the charge distribution varies, we have given a schematic illustration in **Fig. 5c**. It is recommended that you can refer to this schematic illustration when read this description of the phenomena. Moreover, according to your kind suggestion, we have also revised the description of the phenomena to explain more clearly and concretely as far as possible.

Under the built-in electric field of BTO, the lithium ions will redistribute on the LCO/LPSCI/BTO triple-phase interface (TPI). Driven by the Coulomb interaction, both lithium ions in LPSCI initially located behind the BTO (near the side of positive poles of BTO) and lithium ions in LCO originally located across from the BTO (near the side of negative poles of BTO) will migrate toward the vicinities of the LCO/LPSCI/BTO TPI in order to maintain local charge neutralities. Owing to the limited action scope of Coulomb interaction, the amount of lithium ions that migrate to the interface decreases as they move away from the LCO/LPSCI/BTO TPI. Therefore, overall, the lithium-ion-deficient negative charge region on the LPSCI side and the lithium-ion-enriched positive charge region on the LCO side should be significantly restrained. Unfortunately, the result of corresponding charge distribution at 0 V was not obtained by DPC-STEM due to the greater impact from the “edge effect”. As for the LCO/LPSCI interface without bias voltage, much more charges are accumulated at the interface, which can neutralize some effect of false image from “edge effect”. Therefore, the DPC-STEM result of LCO/LPSCI interface can still appear the charge distribution with positive and negative charge separation. By comparison, much less charges are accumulated at the interface after BTO modification. Accordingly, the effect of false image from “edge effect” is particularly obvious. Thereupon, the DPC-STEM result of LCO/LPSCI interface does not appear the obvious charge distribution with

positive and negative charge separation, but appears a false image of only the positive charge layer. Since there is some inaccuracy in the DPC-STEM results without bias voltage, we have moved these results to the supplementary information just for reference because it can still explain the charge distribution of BTO-LCO/LPSCI interface from the side. The corresponding discussions have been revised on page 8 in the manuscript.

Fig. 5 Schematic illustration of interface charge redistribution between oxide cathodes and sulfide SEs before and after different interface engineering. **a**, The initial interface. **b**, The double-phase interface with fast Li-ion conductor CIBLs. **c**, The TPI with discontinuous ferroelectric nanoparticle. The regions with light colors correspond to the relative Li-ion

deficiency. (FMs: ferroelectric materials)

In Fig.4d, only the positive charge layer is discussed. How does the total charge neutrality maintain?

Response: Thank you very much for the good question. As the answers above, the partial differential treatment unavoidably bring about a sudden change at the interface (also called “edge effect”), which will influence the charge distribution at the interface to some extent. As for the LCO/LPSCI interface without bias voltage, much more charges are accumulated at the interface, which can neutralize some effect of false image from “edge effect”. Therefore, the DPC-STEM result of LCO/LPSCI interface can still appear the charge distribution with positive and negative charge separation. By comparison, much less charges are accumulated at the interface after BTO modification. Accordingly, the effect of false image from “edge effect” is particularly obvious. Thereupon, the DPC-STEM result of LCO/LPSCI interface does not appear the obvious charge distribution with positive and negative charge separation, but appears a false image of only the positive charge layer. Since there is some inaccuracy in the DPC-STEM results without bias voltage, we have moved these results to the supplementary information just for reference because it can still explain the charge distribution of BTO-LCO/LPSCI interface from the side. The corresponding discussions have been supplemented on page 8 in the manuscript, and page 10–12 in the supplementary information .

More details of FEM simulations should be described. What chemical potentials of lithium ion were assumed?

Response: Thank you very much for your nice suggestion. According to your suggestion, the more details of FEM simulations have been described on page 15 in the manuscript. Our FEM simulations are based on the semiconductor analogy via the commercial software COMSOL Multiphysics. In this software, the parameters to be assumed include the relative dielectric constant, the band gap, the electron affinity, the state effective density (valence band and conduction band), electron mobility and hole (lithium-ion) mobility. Therefore, the chemical potential of lithium ions was not directly assumed in our FEM simulations. The chemical potential of lithium ions in this work is related to the inherent properties of semiconductor materials. The chemical potential of lithium ions in LCO and LPSCI reported in previous literature^{16,17} is about -4.02 eV and -2 eV, respectively.

In Fig. 6b the total potential difference looks smaller than Fig. 6a. The lithium diffusion may stop when the chemical potential and electrostatic potential is balanced. Why does the lithium diffusion stop in the case of Fig. 6b?

Response: Thank you very much for this question. Takada et al.¹⁸ reported that solid electrolytes will suffer from anodic polarization at the interface when contacting high-voltage cathodes. Since the electrochemical potential of lithium ions ($\tilde{\mu}_{\text{Li}^+} = \mu_{\text{Li}^+} + e\phi$, where $\tilde{\mu}_{\text{Li}^+}$ and μ_{Li^+} are the electrochemical and chemical potential of lithium ions, respectively, and e and ϕ are the elementary charge and the local electrostatic potential, respectively) should be constant across the interface, the anodic polarization increases the electrostatic energy ($e\phi$) and thus decreases μ_{Li^+} on the electrolyte side¹⁹. Additionally, lithium ions are weakly bonded to the anionic framework in sulfide electrolytes, indicating relatively high μ_{Li^+} in the bulk. Therefore, the lithium-ion concentration on the LPSCI side of the interface will decrease, while the lithium-ion concentration on the LCO side of the interface will increase. And as you pointed out, the lithium diffusion may be stopped when the chemical potential and electrostatic potential is balanced. This is also the origin of the internal electrical field from SCL. After introducing a reverse electric field from the polarity of the BTO nanoparticles, the interface lithium ions will be redistributed, which leads to the restrained SCL effect similar to the result of the suppressed lithium diffusion from LPSCI to LCO. Therefore, in **Fig. 6b** the total potential difference looks smaller than **Fig. 6a**. When the chemical potential and electrostatic potential is balanced in the BTO-LCO/LPSCI interface, the lithium diffusion will be also stopped in the case of **Fig. 6b**.

In Fig. 4e-i, it is not clear if the charge density at 0 V is subtracted similarly in Fig. 1.

Response: Thank you very much for your question. In **Fig.4e-i**, the charge density at 0 V is subtracted similarly in **Fig. 1**. The corresponding description has been supplemented on page 8 in the manuscript.

Replies to Reviewer #2:

High interface impedance is one of the bottle neck in the solid state battery technology, and the so-called space charge layers (SCL, also called electrical double layer) might be one of the culprit. Direct microscopy imaging of the SCL effect has hitherto not possible. By using the latest imaging technology called differential phase contrast – scanning transmission electron microscopy (DPC-STEM), the authors demonstrate the possibility of visualizing the SCL effect directly.

Furthermore, they show that introducing built-in electric field on the cathode/electrolyte interface, it is possible to suppress the formation of SCL, thus benefitting the interface ion transport. The report constitutes a timely new contribution to the solid state battery research field, and it advances significantly our understanding to the SCL effect on the impedance of solid state batteries. The paper is well written, and the quality is high. I thus recommend publication of the manuscript in Nature Communications after the authors address the following minor issues.

Response: Thank you very much for your positive comments and recommending publication of our manuscript in Nature Communications. In the revised manuscript, we have addressed all the minor issues that you have asked. The revisions are highlighted with yellow in the manuscript and supplementary information.

1. The sulfide solid state electrolyte is extremely sensitive to moisture. The authors did not describe in detail how they fabricate thin foil TEM samples from a bulk solid state battery. Neither do they provide details regarding how the samples are transferred from air to FIB, and then from FIB to TEM sample holder, and how the holder was inserted into the TEM without air exposure. They do show that by coating Al_2O_3 onto the sample, air damage to the sample was avoided. But how was this done and at what stage? Any air exposure for even a nanosecond renders the SE useless. The air sensitivity of sulfide electrolyte is actually what impedes the application of in-situ TEM in this important research field.

Response: Thank you very much for your helpful suggestions. As you pointed out, the sulfide solid-state electrolyte is extremely sensitive to moisture, so we have protected the samples from direct air exposure at every experimental stage. After the bulk solid-state battery without anode was prepared in an Ar-filled dry glovebox, the solid-state battery pellet was sealed in the Ar-filled chamber of a special sample stage in the glovebox. Then, the stage was taken out from the glovebox and transferred to the FIB-SEM system (FEI Helios Nanolab 460HP). The relatively flat place of solid-state battery pellet in the microscopic field of vision was selected as the further processing area. Firstly, the thin Pt protective layer was deposited on the selective area under the electron beam with relatively low energy. After that, the selective specimen ($10 \times 1.5 \times 1 \mu\text{m}$) was taken and placed across the middle (observation area) of the chip. And then, the specimen was fixed and connected to the Pt electrode at both ends by redepositing a thin Pt layer. Finally, the specimen in the observation area was thinned by focusing ion beam (FIB). The fabricated in-situ

solid-state battery chip was sealed in the chamber of the special sample stage (the chamber was vacuum at present). Then, the stage was taken out from the FIB-SEM system and transferred into another Ar-filled dry glovebox. In the glovebox, the chip was taken out from the stage and sealed in the Ar-filled chamber of another special sample stage of an ALD system. After coating ALD Al_2O_3 onto the chip, the chip was put on the DENS single tilt specimen holder to carry out the *in-situ* DPC-STEM experiments. In conclusion, there is no any air exposure for our specimen during the whole preparation and transfer process, which can avoid the sulfide SE useless. The corresponding details have been supplemented on page 12 in the manuscript.

2. Based on Fig. 2 in ref. 20, upon biasing the battery, the potential in the LCO side (essentially caused by the changes of Li ion concentration) changes with the biasing voltage accordingly, however, such changes are not reflected in the DPC imaging. Can the authors provide an explanation for this?

Response: Thank you very much for this question. The DPC imaging can provide the electric-field distribution, while the electron holography (EH) imaging (ref. 20) can give the potential distribution. Essentially, as you pointed out, the change of electric field and potential with the biasing voltage should be consistent, because these variations are both resulted from the changes of Li ion concentration caused by lithium-ion diffusion. The net electric-field distribution at the LCO/LPSCI interface obtained by DPC imaging is shown in **Fig. R1**. It can be found that the electric-field in the LCO side also changes with the biasing voltage accordingly. However, what we concerned about in this manuscript is the interfacial charge distribution and accumulation resulting from SCL. So we further treated the electric field with partial differential to obtain the charge density. As a result, the region with larger variations in the DPC results of electric-field distribution changes more pronouncedly in the DPC results of net charge distribution, while the region with smaller variations in the DPC results of electric-field distribution changes more slightly in the DPC results of net charge distribution. Therefore, the change of net charge at the interface is more obvious, while the change of net charge inside the bulk is less obvious. However, the electric-field distribution in our DPC imaging is slightly different from the potential distribution in the EH imaging (Fig. 2 in ref. 20). This is probably because the experimentally observed potential on Fig. 2 in ref. 20 is affected by the electric field leaking from the sample,^{3,20} since they did not use the corresponding electric shielding films (*e.g.* ALD Al_2O_3).^{20,21}

Fig. R1 In-situ DPC-STEM observations of net electric-field distribution (subtracting the corresponding result without bias voltage) at the LCO/LPSCI interface with the bias voltage of 1.2, 1.4, 1.6, 1.8, 2.0 and 2.2 V.

3. Is it possible to provide a quantitative measurement of the width of the SCL layer? And how does it vary with biasing? How high is the voltage and how strong is the field produced by the SCL?

Response: Thank you very much for the good questions. The three questions you raised here are actually key scientific issues and challenges to further advance SCL evaluation in ASSLIBs, and so far there have been relatively few studies on these areas. Regarding to the quantitative measurement of the width of the SCL layer in ASSLIBs, Takada et al.²² firstly provided an estimated thickness of LCO/Li_{3.25}Ge_{0.25}P_{0.75}S₄ and LiMn₂O₄/Li_{3.25}Ge_{0.25}P_{0.75}S₄ interface is about 10 nm based on the analysis of charge curves. Recently, Nomura et al.²⁰ have studied the Cu/Li_{1+x+y}Al_x(Ti,Ge)_{2-x}Si_yP_{3-y}O₁₂ (LASGTP) interface by the phase-shifting electron holography technique and shown that the thickness of the SCL was 10 nm. Moreover, they also concluded that the maximum potential height of the SCL was 1.3 V. Although it is an important advancement for the quantitative measurement of the SCL, it may not actually reflect the true state of the SCL because they did not consider the effects of the MIP and possible dynamical diffraction as the first reviewer pointed out. From our DPC results (**Fig. 1** and **Fig. R1**) of LCO/LPSCI interface, it can

be found that the plausible width of the SCL is about 10–20 nm at the lower bias voltage (e.g. 1.0, 1.2 V). With the increasing of the bias voltage, the width and electric-field seem to get larger. Specially, width expansion to the cathode side may be more pronounced than expansion to the electrolyte side. Nevertheless, based on the existing DPC imaging technique, it is still difficult to accurately identify the width and electric-field of the SCL evolving with the bias voltage. Therefore, the key point of our manuscript has been put on the interfacial charge distribution and accumulation resulting from SCL. Anyway, we will keep an eye on your good questions and try to make new progress in future.

Fig. 1 *In-situ* charge-distribution characterization of LCO/LPSCI interface. **a–c**, The HADDF-STEM image of LCO/LPSCI interface and corresponding mappings of Co and S element. **d–i**, *In-situ* DPC-STEM observations of net charge accumulation at the LCO/LPSCI interface with the bias voltage of 1.0, 1.2, 1.4, 1.6, 2.0 and 2.2 V.

Fig. R1 In-situ DPC-STEM observations of net electric-field distribution (subtracting the corresponding result without bias voltage) at the LCO/LPSCI interface with the bias voltage of 1.2, 1.4, 1.6, 1.8, 2.0 and 2.2 V.

References

1. Migunov, V., London, A., Farle, M. & Dunin-Borkowski, R. E. Model-independent measurement of the charge density distribution along an Fe atom probe needle using off-axis electron holography without mean inner potential effects. *J. Appl. Phys.* **117**, 134301 (2015).
2. Yao, Y. et al. In situ electron holography study of charge distribution in high-kappa charge-trapping memory. *Nat. Commun.* **4**, 2764 (2013).
3. Aizawa, Y. et al. In situ electron holography of electric potentials inside a solid-state electrolyte: Effect of electric-field leakage. *Ultramicroscopy* **178**, 20–26, (2017).
4. Cassidy, C., Dhar, A. & Shintake, T. Determination of the mean inner potential of cadmium telluride via electron holography. *Appl. Phys. Lett.* **110**, 163503 (2017).
5. Yesibolati, M. N. et al. Mean Inner Potential of Liquid Water. *Phys. Rev. Lett.* **124**, 065502 (2020).
6. Beleggia, M., Gontard, L. C. & Dunin-Borkowski, R. E. Local charge measurement using off-axis electron holography. *J. Phys. D Appl. Phys.* **49**, 294003 (2016).
7. Gan, Z., DiNezza, M., Zhang, Y. H., Smith, D. J. & McCartney, M. R. Determination of Mean Inner Potential and Inelastic Mean Free Path of ZnTe Using Off-Axis Electron Holography and Dynamical Effects Affecting Phase Determination. *Microsc. Microanal.* **21**, 1406–1412 (2015).
8. Zheng, C. L., Scheerschmidt, K., Kirmse, H., Hausler, I. & Neumann, W. Imaging of three-dimensional (Si,Ge) nanostructures by off-axis electron holography. *Ultramicroscopy* **124**, 108–116 (2013).
9. Ding, Y. et al. Quantifying mean inner potential of ZnO nanowires by off-axis electron holography. *Micron* **78**, 67–72 (2015).
10. Lee, G. et al. Influence of combinatory effects of STEM setups on the sensitivity of differential phase contrast imaging. *Micron* **127**, 102755 (2019).
11. Majert, S. & Kohl, H. High-resolution STEM imaging with a quadrant detector--conditions for differential phase contrast microscopy in the weak phase object approximation. *Ultramicroscopy* **148**, 81–86 (2015).
12. Bauer, B. et al. Direct detection of spontaneous polarization in wurtzite GaAs nanowires. *Appl. Phys. Lett.* **104**, 211902 (2014).
13. Zhu, X. N., Chen, X., Tian, H. & Chen, X. M. Atomic scale investigation of enhanced ferroelectricity in (Ba,Ca)TiO₃. *RSC Advances* **7**, 22587–22591 (2017).
14. Zhu, D. P., Mangeri, J., Wang, R. L. & Nakhmanson, S. Size, shape, and orientation dependence of the field-induced behavior in ferroelectric nanoparticles. *J. Appl. Phys.* **125**, 134102 (2019).
15. Sato, Y. et al. Atomic-Scale Observation of Titanium-Ion Shifts in Barium Titanate Nanoparticles: Implications for Ferroelectric Applications. *ACS Appl. Nano Mater.* **2**, 5761–5768 (2019).
16. Swift, M. W. & Qi, Y. First-Principles Prediction of Potentials and Space-Charge Layers in All-Solid-State Batteries. *Phys. Rev. Lett.* **122**, 167701 (2019).
17. Zhu, Y., He, X. & Mo, Y. Origin of Outstanding Stability in the Lithium Solid Electrolyte Materials: Insights from Thermodynamic Analyses Based on First-Principles Calculations. *ACS Appl. Mater. Interfaces* **7**, 23685–23693 (2015).
18. Takada, K., Ohno, T., Ohta, N., Ohnishi, T. & Tanaka, Y. Positive and Negative Aspects of

- Interfaces in Solid-State Batteries. *ACS Energy Lett.* **3**, 98–103 (2017).
19. Maier, J. Ionic-Conduction in-Space Charge Regions. *Prog. Solid State Ch.* **23**, 171–263 (1995).
 20. Nomura, Y. et al. Direct Observation of a Li-Ionic Space-Charge Layer Formed at an Electrode/Solid-Electrolyte Interface. *Angew. Chem. Int. Ed.* **58**, 5292–5296 (2019).
 21. Nomura, Y., Yamamoto, K., Hirayama, T. & Saitoh, K. Electric shielding films for biased TEM samples and their application to in situ electron holography. *Microscopy* **67**, 178–186 (2018).
 22. Takada, K. et al. Interfacial phenomena in solid-state lithium battery with sulfide solid electrolyte. *Solid State Ionics* **225**, 594–597 (2012).

Reviewers' Comments:

Reviewer #1:

Remarks to the Author:

After reading the response letter, I still confused about the authors' explanation. In DPC STEM, what we initially obtain should be the electric field map, that is the differential of the electrostatic potential. This is different from electron holography. Electron holography visualizes electrostatic potential. So, what the color contrast in Fig1? I thought it should be related to in-plane electric field vector but there is no such color wheel. If this is the total charge density map obtained by differentiating the electric field vector map obtained by DPC STEM, it should be well documented.

The MIP difference at the heterointerface causes built-in electric field. Because of this, further differentiation of DPC images also causes contrast at the interface due to the MIP difference. Therefore, when we observe the charge density map of the heterointerface by differentiating DPC, the image should be the superposition of the charges due to the built-in electric field (MIP difference) and the charges due to the redistribution of ions and electrons at the interface. The authors obviously would like to visualize the latter, but the former is always there. Therefore, we should extract the latter from the images somehow. This is what I said the effect of MIP. The authors say in the response, "The partial differential result of the equal electric field is zero. As a result, the effect of MIP on the charge distribution can be eliminated in the same material." If there is no MIP differences, the differentiation will eliminate the electric field effect. BUT, at the interface, there must be the MIP difference and the resultant built-in electric field. Supplementary Fig10 is very confusing. MIP does not cause the electric field in the constant thickness, but MIP gradient causes the electric field. So what the authors called "edge effect" is the built-in electric field due to the difference in MIP. This electric field component due to the MIP differences should be removed from the DPC images in order to discuss the true SCL structures. Therefore, this point must be cleared.

Reviewer #2:

Remarks to the Author:

All my questions have been clearly addressed by the authors. I recommend publication in its present form.

Point-by-point responses to the reviewers' comments

Replies to Reviewer #1:

After reading the response letter, I still confused about the authors' explanation. In DPC STEM, what we initially obtain should be the electric field map, that is the differential of the electrostatic potential. This is different from electron holography. Electron holography visualizes electrostatic potential. So, what the color contrast in Fig. 1? I thought it should be related to in-plane electric field vector but there is no such color wheel. If this is the total charge density map obtained by differentiating the electric field vector map obtained by DPC STEM, it should be well documented.

Response: Thank you very much for your question. As you mentioned above, in DPC-STEM measurements, what we initially obtain is exactly the electric field map. In the following we will detail what the color contrast in **Fig. 1**. In **Fig. 1**, the color contrast of LCO/LPSCI interface with bias voltage is the net charge density map obtained by differentiating the net electric field, instead of the total charge density map by differentiating the electric field directly detected by DPC-STEM. As shown in **DPC characterization of Methods** (on page 15, 16 of manuscript), the electric field obtained from DPC-STEM includes the components due to the SCL, mean inner potential (MIP) difference, and possible dynamical diffraction effect (DDE). The electric fields caused by the MIP difference and DDE are constant after cathode/electrolyte contact, while the electric field caused by the SCL varies with bias voltage. However, it is impossible to separate the individual role of SCL and MIP electric field on interfacial lithium-ion transport during battery working recently. As reported previously, the electric field component due to the MIP difference can be removed from the DPC electric field images at different bias voltages by subtracting the corresponding electric-field result at 0 V before the partial differential treatment (described in detail on page 15, 16 of manuscript).¹⁻³ So, in order to study the true SCL effect on interfacial lithium-ion transport, we investigate the interface net electric field, in which the electric field component due to the MIP difference has been removed from the DPC electric field images at different bias voltages. **Supplementary Fig. 6** shows the corresponding net electric field map of **Fig. 1** at the LCO/LPSCI interface with different bias voltages, where colorful arrows denote the directions and strengths of the electric fields. The software used to draw the vector maps is Avizo. According to the Gauss's law, the charge density map can be obtained by calculating the

differential of the electric field map measured by DPC-STEM.⁴⁻⁹ Therefore, the color contrast in **Fig. 1** is the net charge density map by differentiating the net electric field shown in **Supplementary Fig. 6**. The corresponding revisions have been supplemented on page 7 in the supplementary information.

Actually, some pioneers have used the DPC-STEM technique to map the charge density profile of the InGaN/GaN superlattice with 2.2-nm In_{0.15}Ga_{0.85}N quantum wells,⁴ SrTiO₃/BiFeO₃ heterojunction,⁶ GaN thin crystalline,⁷ SrTiO₃ single crystal,⁸ and wurtzite GaAs nanowires.⁹ Additionally, according to the Poisson's equation, the charge density map can be also obtained by calculating the second differential of the electrostatic potential map obtained by electron holography TEM (EH-TEM).^{2,10-13} However, compared to the EH-TEM, the DPC-STEM can provide a field of view in the order of hundreds of nanometers, which is not easily achieved in EH-TEM.^{8,14,15} Therefore, in this manuscript, we use the advanced DPC-STEM technique to directly observe the electrode/electrolyte interface lithium-ion accumulation resulting from SCL by investigating the net charge density distribution across the interface.

Supplementary Figure 6. *In-situ* DPC-STEM observations of net electric field distribution (subtracting the corresponding result without bias voltage) at the LCO/LPSCI interface with the bias voltage of 1.0, 1.2, 1.4, 1.6, 2.0 and 2.2 V.

The MIP difference at the heterointerface causes built-in electric field. Because of this, further

differentiation of DPC images also causes contrast at the interface due to the MIP difference. Therefore, when we observe the charge density map of the heterointerface by differentiating DPC, the image should be the superposition of the charges due to the built-in electric field (MIP difference) and the charges due to the redistribution of ions and electrons at the interface. The authors obviously would like to visualize the latter, but the former is always there. Therefore, we should extract the latter from the images somehow. This is what I said the effect of MIP. The authors say in the response, “The partial differential result of the equal electric field is zero. As a result, the effect of MIP on the charge distribution can be eliminated in the same material.” If there is no MIP differences, the differentiation will eliminate the electric field effect. BUT, at the interface, there must be the MIP difference and the resultant built-in electric field. Supplementary Fig. 10 is very confusing. MIP does not cause the electric field in the constant thickness, but MIP gradient causes the electric field. So what the authors called “edge effect” is the built-in electric field due to the difference in MIP. This electric field component due to the MIP differences should be removed from the DPC images in order to discuss the true SCL structures. Therefore, this point must be cleared.

Response: Thank you very much for your kind suggestion. We agree with you that the charge density map of the heterointerface by differentiating DPC is the superposition effect of the MIP and SCL. But, it should be pointed out that what we focus on in this manuscript is the effect of SCL on interfacial lithium-ion transport in the working all-solid-state batteries. And actually, as mentioned in the above response, the electric field component due to the MIP difference has been removed from the DPC electric field images at different bias voltages by subtracting the corresponding electric-field result at 0 V before the partial differential treatment.¹⁻³ Therefore, the net charge density distribution results in the DPC images at different bias voltages exactly reflect the effect of the true SCL, which is enough to support our conclusion.

Of course, in order to better understand the pristine charge density distribution from the true SCL, this electric field component due to the MIP difference should be best removed from the DPC electric field images at 0 V before the partial differential treatment. However, although we can obtain the pristine electric field of cathode and electrolyte respectively before their contact, the electric field component due to the MIP difference at 0 V can't be removed accurately by subtracting because the electric field due to the MIP difference at the interface has changed after

cathode/electrolyte contact, accompanied by the electric field generated by the migration of Li ions (*i.e.* SCL). Therefore, in **Supplementary Fig. 10** (last submitted supplementary information), the schematic illustration of the “edge effect” on the DPC result of the charge density distribution at the interface without bias voltage is really not accurate, because the electric field caused by MIP of LCO and LPSCI were assumed as a constant respectively without considering the change of interface electric field after cathode/electrolyte contact. As a result, we have deleted this figure in the supplementary information. However, there is no doubt that the built-in electric field due to the MIP difference does interfere the DPC-STEM result (only from the SCL) at 0 V to some extent. Unfortunately, to the best of our knowledge, so far there have been no any feasible methods to remove the electric field component due to the MIP difference from the DPC images at 0 V. As a result, this is also why the previous studies with *in-situ* EH-TEM or DPC-STEM didn’t show the result at 0 V.^{1-3,16}

The Li element mapping obtained by EELS can reflect the migration of Li ions after cathode/electrolyte contact,¹⁷⁻¹⁹ so we further analyze the Li and Co elemental profiles from the EELS line scan (**Supplementary Fig. 8**) at 0 V on the interface. On the cathode side, it can be found that interfacial lithium ions are more than bulk lithium ions before BTO coating (**Supplementary Fig. 8a,c**). This indicates that there appears obvious lithium-ion enrichment on the LCO side at the interface due to the lithium-ion diffusion from electrolyte to cathode. However, this lithium-ion enrichment on the LCO side is clearly suppressed after BTO coating (**Supplementary Fig. 8b,d**). When combined with different DPC-STEM results of LCO/LPSCI (the charge density distribution with positive and negative charge separation) and BTO-LCO/LPSCI (the false image of only the positive charge layer) interfaces, it can be inferred that at the interface without BTO coating, the lithium-ion enrichment on the LCO side should be the main source for the positive charge density distribution, while the corresponding lithium vacancies should be the main source for the negative charge density distribution. This is because much more charges are accumulated at the interface, which can neutralize the effect of false image from the MIP difference. Therefore, the DPC-STEM result of LCO/LPSCI interface can still appear the charge density distribution with positive and negative charge separation. By comparison, much less charges accumulate at the interface after BTO modification. Accordingly, the effect of false image from the MIP difference is particularly obvious. Thereupon, the

DPC-STEM result of LCO/LPSCI interface does not appear the obvious charge density distribution with positive and negative charge separation, but appears a false image of only the positive charge density layer. Since there is some inaccuracy in the DPC-STEM results without bias voltage, we have moved these results to the supplementary information just for reference because it can still explain the charge density distribution of LCO/LPSCI and BTO-LCO/LPSCI interface from the side. The corresponding discussions have been supplemented on page 4, 5, 9, 15, 16 in the manuscript, and page 9 in the supplementary information.

Supplementary Figure 8. The HADDF-STEM images of LCO/LPSCI and BTO-LCO/LPSCI interface, and the corresponding EELS line scan of Co and Li.

References

1. Migunov, V., London, A., Farle, M. & Dunin-Borkowski, R. E. Model-independent measurement of the charge density distribution along an Fe atom probe needle using off-axis electron holography without mean inner potential effects. *J. Appl. Phys.* **117**, 134301 (2015).
2. Yao, Y. et al. In situ electron holography study of charge distribution in high-kappa charge-trapping memory. *Nat. Commun.* **4**, 2764 (2013).
3. Aizawa, Y. et al. In situ electron holography of electric potentials inside a solid-state electrolyte: Effect of electric-field leakage. *Ultramicroscopy* **178**, 20–26, (2017).
4. Haas, B., Rouviere, J. L., Boureau, V., Berthier, R. & Cooper, D. Direct comparison of off-axis holography and differential phase contrast for the mapping of electric fields in semiconductors by transmission electron microscopy. *Ultramicroscopy* **198**, 58–72 (2019).
5. Muller, K. et al. Atomic electric fields revealed by a quantum mechanical approach to electron picodiffraction. *Nat. Commun.* **5**, 5653 (2014).
6. Gao, W. et al. Real-space charge-density imaging with sub-angstrom resolution by four-dimensional electron microscopy. *Nature* **575**, 480–484 (2019).
7. Sanchez-Santolino, G., et al. Probing the Internal Atomic Charge Density Distributions in Real Space. *ACS Nano* **12**, 8875–8881 (2018).
8. Shibata, N. et al. Electric field imaging of single atoms. *Nat. Commun.* **8**, 15631 (2017).
9. Bauer, B., Hubmann, J., Lohr, M., Reiger, E., Bougeard, D. & Zweck, J. Direct detection of spontaneous polarization in wurtzite GaAs nanowires. *Appl. Phys. Lett.* **104**, 211902 (2014).
10. Song, K. et al. Correlative High-Resolution Mapping of Strain and Charge Density in a Strained Piezoelectric Multilayer. *Adv. Mater. Interfaces* **2**, 1400281 (2015).
11. Wu, Z. H. et al. Mapping the electrostatic potential across AlGaN/AlN/GaN heterostructures using electron holography. *Appl. Phys. Lett.* **90**, 032101 (2007).
12. Wu, Z. H. et al. Effect of internal electrostatic fields in InGaN quantum wells on the properties of green light emitting diodes. *Appl. Phys. Lett.* **91**, 041915 (2007).
13. Cai, J. & Ponce, F. A. Study of charge distribution across interfaces in GaN/InGaN/GaN single quantum wells using electron holography. *J. Appl. Phys.* **91**, 9856–9862 (2002).
14. Zhang, C., Feng, Y., Han, Z., Gao, S., Wang, M. & Wang, P. Electrochemical and Structural Analysis in All-Solid-State Lithium Batteries by Analytical Electron Microscopy: Progress and Perspectives. *Adv. Mater.*, e1903747 (2019).
15. Lohr, M. et al. Differential phase contrast 2.0—opening new "fields" for an established technique. *Ultramicroscopy* **117**, 7–14 (2012).
16. Yamamoto, K. et al. Dynamic visualization of the electric potential in an all-solid-state rechargeable lithium battery. *Angew. Chem. Int. Ed.* **49**, 4414–4417 (2010).
17. Nomura, Y., Yamamoto, K., Hirayama, T., Ouchi, S., Igaki, E. & Saitoh K. Direct Observation of a Li-Ionic Space-Charge Layer Formed at an Electrode/Solid-Electrolyte Interface. *Angew. Chem. Int. Ed.* **58**, 5292–5296 (2019).
18. Nomura, Y., Yamamoto, K., Hirayama, T., Igaki, E. & Saitoh, K. Visualization of Lithium Transfer Resistance in Secondary Particle Cathodes of Bulk-Type Solid-State Batteries. *ACS Energy Lett.* **5**, 2098–2105 (2020).
19. Nomura, Y. et al. Quantitative Operando Visualization of Electrochemical Reactions and Li Ions in All-Solid-State Batteries by STEM-EELS with Hyperspectral Image Analyses. *Nano Lett.* **18**, 5892–5898 (2018).